# Fever integrates antimicrobial defences, inflammation control, and tissue repair in a cold-blooded vertebrate

**Farah Haddad[1†], Amro M Soliman[1†], Michael E Wong[1], Emilie H Albers[1], Shawna L Semple[1], Débora Torrealba[1], Ryan D Heimroth[2], Asif Nashiry[1], Keith B Tierney[1], Daniel R Barreda[1]\***

[1]University of Alberta, Edmonton, Canada; [2]Emory University, Atlanta, United States

**\*For correspondence:**
d.barreda@ualberta.ca

[†]These authors contributed equally to this work

**Competing interest:** The authors declare that no competing interests exist.

**Abstract** Multiple lines of evidence support the value of moderate fever to host survival, but the mechanisms involved remain unclear. This is difficult to establish in warm-blooded animal models, given the strict programmes controlling core body temperature and the physiological stress that results from their disruption. Thus, we took advantage of a cold-blooded teleost fish that offered natural kinetics for the induction and regulation of fever and a broad range of tolerated temperatures. A custom swim chamber, coupled to high-fidelity quantitative positional tracking, showed remarkable consistency in fish behaviours and defined the febrile window. Animals exerting fever engaged pyrogenic cytokine gene programmes in the central nervous system, increased efficiency of leukocyte recruitment into the immune challenge site, and markedly improved pathogen clearance in vivo, even when an infecting bacterium grew better at higher temperatures. Contrary to earlier speculations for global upregulation of immunity, we identified selectivity in the protective immune mechanisms activated through fever. Fever then inhibited inflammation and markedly improved wound repair. Artificial mechanical hyperthermia, often used as a model of fever, recapitulated some but not all benefits achieved through natural host-driven dynamic thermoregulation. Together, our results define fever as an integrative host response that regulates induction and resolution of acute inflammation, and demonstrate that this integrative strategy emerged prior to endothermy during evolution.

## Editor's evaluation

This study makes two important advances: First, the authors developed a new experimental system to study behavioral control of body temperature in fish. Second, using this experimental paradigm, the authors uncover the impact of body temperature regulation on immune defense and tissue repair. It presents important new insights into conserved defense mechanisms and as such the study would be of broad interest.

## Introduction

Fever is a cornerstone of acute inflammation (*Rosenberg and Gallin, 1999*). The classical response is initiated through the recognition of damage-associated molecular patterns or pathogen-associated molecular patterns by pattern recognition receptors on the surface of immune leukocytes. This sensing reaction leads to the activation of resident myeloid cells at the challenge site, and is followed by rapid production of pyrogenic prostaglandin E2 ($PGE_2$) and cytokines, including tumor necrosis factor alpha (TNFA), interleukin-1 beta (IL1B), and interleukin-6 (IL-6) (*Engel et al., 1994*). Contributions at multiple levels of the fever cascade are observed; IL-6, for example, promotes initial activation events,

the early rise in core body temperature, as well as the subsequent orchestration of lymphocyte trafficking to lymphoid organs (*Evans et al., 2015*). Others, like interleukin-8 (IL-8) are produced locally at the site of infection and manage recruitment of inflammatory leukocytes from circulation, peripheral stores, and the hematopoietic compartment (*Deniset and Kubes, 2018*). Increased synthesis of cyclooxygenase 2 (COX-2) within the median preoptic nucleus of the hypothalamus fosters added production of $PGE_2$, which serves a dominant pyrogenic role in fever (*Cao et al., 1996*). Production of endogenous pyrogenic cytokines such as IL1B and IL-6 within the central nervous system (CNS) may also complement the activity of those generated in peripheral tissues during the induction of fever (*Evans et al., 2015*). Systemic physiological changes in vasodilation, vascular permeability, and leukocyte recruitment become evident a few hours after the initial insult (*Mackowiak, 1998*). Thus, early innate immune recognition initiates a highly organized response that engages neuronal circuitry in the central and peripheral nervous systems and triggers the activation of thermoregulatory pathways.

The association of fever and disease dates back at least as far as Hippocrates (2500 years) (*Atkins, 1985*). A rise in body temperature is so tightly associated with the inflammatory response that heat (*calor*) is one of the four cardinal signs of inflammation. While non-severe forms of fever are well established to increase host survival upon infection (*Evans et al., 2015*; *Covert and Reynolds, 1977*; *Kluger et al., 1975*), the mechanisms behind these contributions remain poorly understood. Improved host protection has been postulated to stem from a direct impact of increased temperature on invading pathogens, and global upregulation of antimicrobial immune mechanisms (*Evans et al., 2015*; *Casadevall, 2016*). For thermal restriction, pathogen survival and replication can be directly compromised when their maximum tolerated temperatures are reached or exceeded (*Casadevall, 2016*). This is well documented for many microbes and served as an effective therapy against diseases like neurosyphilis and gonorrhea prior to the advent of antibiotics (*Kluger et al., 1975*). However, many pathogens are known to be largely unaffected or grow better at the higher temperatures that fever elicits (*Casadevall, 2016*; *Shapiro and Cowen, 2012*). And, even for originally permissive organisms, the effectiveness of thermal restriction may be limited to initial encounters between pathogen and host, given the extensive repertoire of thermal resistance mechanisms that are available to viruses, archaea, bacteria, fungi, and parasites (*Casadevall, 2016*; *Shapiro and Cowen, 2012*). Global promotion of immune defenses has also been proposed, based on reported effects of temperature increases on metabolic rates and multiple effectors and regulators of innate and adaptive immunity (*Evans et al., 2015*; *Bennett and Nicasrti, 1960*). However, the global nature of this induction contradicts the emphasis that a host is known to place on energy conservation and management of collateral inflammation-associated tissue damage (*Steiner and Romanovsky, 2019*; *Wang and Medzhitov, 2019*). As a result, debates on the net value of fever to host health continue to permeate the literature (*Atkins, 1985*; *Greisman and Mackowiak, 2002*; *Bernheim and Kluger, 1976*; *Wrotek et al., 2021*; *Nielsen et al., 2013*). This is compounded by limitations among available experimental models to adequately recapitulate the natural physiological processes driving and sustaining febrile responses. Early assessments into the benefits of fever, for example, suffered from temporal deviations – fever was artificially induced prior to infection (*Kluger et al., 1975*; *Bennett and Nicasrti, 1960*). In other cases, thermal ranges outside those normally elicited by fever were used, or peak temperatures were maintained for extended periods of time (*Kluger et al., 1975*; *Bennett and Nicasrti, 1960*). In vitro and in vivo mammalian models of fever-range hyperthermia (FRH) continue to offer valuable insights and have confirmed improvements in host survival and decreased microbial loads because of increased core body temperatures (*Evans et al., 2015*; *Hasday and Singh, 2000*). Unfortunately, exogenous mechanical temperature manipulation is also well established to cause physiological stress and fails to replicate the host's intrinsic thermoregulatory machinery for heating and cooling elicited during natural fever (*Bernheim and Kluger, 1976*). Pharmacological models based on antipyretic drug administration (e.g., nonsteroidal anti-inflammatory drugs [NSAIDs]) have also been used broadly, but are hampered by inhibition of inflammatory pathways at multiple points and other off-target effects (*Bernheim and Kluger, 1976*; *Earn et al., 2014*). As a result, fever remains among the most poorly understood of the acute inflammatory processes.

Ectotherms (fish, amphibians, reptiles, invertebrates) and endotherms (mammals, birds) induce fever upon infection, and both exhibit a strong behavioural component (*Terrien et al., 2011*; *Garami et al., 2018*). Ectotherms, however, rely on behaviour to induce fever in the absence of the metabolic toolkit available to endotherms (*Evans et al., 2015*; *Kluger, 1979*; *Hasday et al., 2014*). Upon infection,

fish move to areas with warmer waters, while reptiles lay on sun-warmed terrestrial environments (*Kluger, 1979*). Social animals like honeybees go further, displaying behavioural thermoregulation at the group level, to achieve a communal increase in hive temperature in response to infection (*Starks et al., 2000*). Despite different physiologies and thermoregulatory strategies, common biochemical pathways appear to regulate fever across cold- and warm-blooded vertebrates (*Evans et al., 2015*; *Boltaña et al., 2013*). In all, the conservation of fever across phylogeny spans 550 million years of metazoan evolution (*Kluger, 1979*). The net result is a survival advantage (*Evans et al., 2015*; *Earn et al., 2014*) which, based on the long-standing natural selection of the fever response, appears to heavily out-weight the reported metabolic costs (*Kluger, 1979*; *Muchlinski, 1985*), increased potential for predation (*Otti et al., 2012*), and decrease in reproductive success (*Graham et al., 2017*). This level of commitment mirrors that displayed by some pathogens to inhibit it. Herpesvirus, for example, has been recently shown to express soluble decoy TNF receptors during infection that delay behavioural fever and allow for increased viral replication (*Rakus et al., 2017*).

In this study, we used a cold-blooded teleost vertebrate model to gain additional insights into the immunobiology of fever. We examined febrile responses under host-driven dynamic thermoregulation, to more closely mimic natural conditions for heating and cooling. This avoided common caveats encountered with exogenous drugs, temporal deviations from native thermoregulatory programmes, or forcing animals beyond thermal ranges normally elicited through fever. An in vivo Aeromonas cutaneous infection model was tailored to focus on the most common moderate self-resolving form of this natural biological process rather than severe pathological fever. Under the experimental conditions used, fever was transient and self-limiting, enabling us to interrogate its potential contributions during induction and resolution phases of acute inflammation. Together, our results demonstrate that fever is not a byproduct of acute inflammation but an important regulator of its induction and control. Fever in ectotherms favours early and selective induction of innate antimicrobial programmes against infection coupled with controlled inflammation and accelerated wound repair. As such, it serves as a fine-tuning mechanism of innate immunity to pathogens.

## Results
### High-resolution motion tracking identifies a predictable fever programme in an outbred population of fish

Conventional shuttle box approaches for examination of fish behaviour are well established to produce significant variability between individual animals, driven by differences in their preference for cover, swim depth and activity level, in addition to social behaviours associated with schooling, territorial grouping or avoidance based on dominance (*Brown et al., 2011*). In an effort to decrease this heterogeneity and offer greater analytical depth to our behavioural outputs, we customized an annular temperature preference tank (ATPT) (*Myrick et al., 2004*) which takes advantage of fluid dynamics instead of physical barriers to establish distinct temperature environments (*Figure 1*). Under this setup, aquatic animals were free to choose environmental temperatures independent of other variables such as lighting, perceived cover, edge effects, water depth, or current that impact positional behaviour. Distinct temperature set points (16, 19, 21, 23, and 26°C) were chosen and used to create a barrier-free environmental housing temperature gradient that spanned a 10°C range (*Figure 1A–C*). Water inputs and flow rates were optimized to yield a gradient that was stable through 14 days of continuous evaluation (*Figure 1C*). Directional flow rates across the swim chamber were also adjusted to create smaller consistent temperature gradients between each of the primary thermal zones (*Figure 1B*). Our goal was to avoid abrupt housing water temperature boundaries, which could affect an animal's choice to transition between adjacent thermal zones. Next, we coupled this updated ATPT to an automated monitoring system with per-second temporal resolution for effective tracking of fish through day and night cycles (*Figure 1D*). This offered greater analytical robustness and temporal resolution than previously possible in behavioural fever analyses. We then challenged individual teleost fish (*Carassius auratus*; goldfish) with an *Aeromonas* cutaneous infection in vivo. As eurythermal ectotherms, goldfish offered an opportunity to examine absolute changes to thermopreference to an immune challenge, while minimizing the potential for thermal stress. This is because the natural range of tolerated environmental temperatures for these fish (1.3–34.5°C) was broader than those temperatures expected in a febrile response. Our choice also provided access to a previously optimized in vivo self-resolving

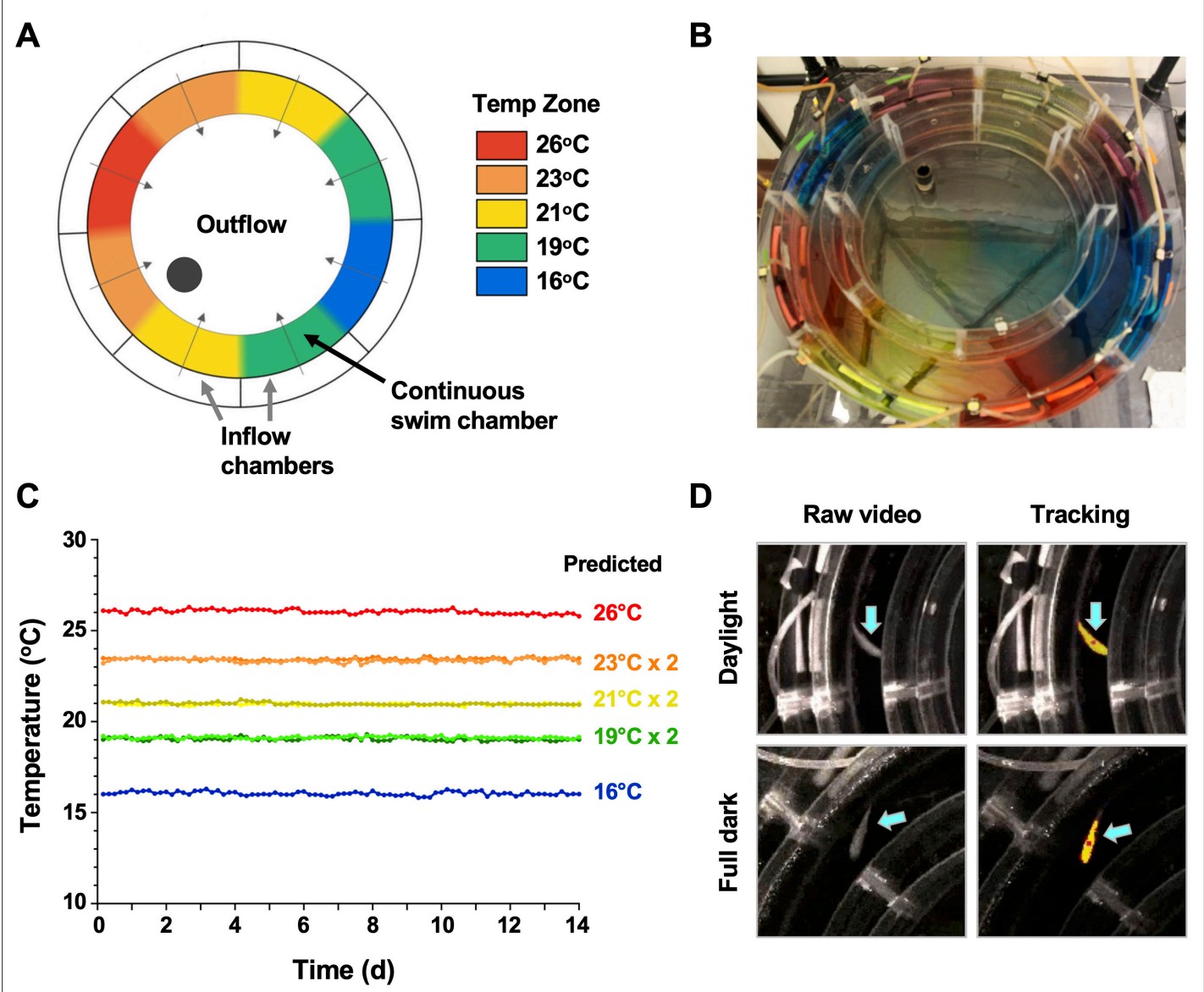

**Figure 1.** Annular temperature preference tank (ATPT) design, validation, and fish tracking. (**A**) The ATPT established a continuous ring-shaped swim chamber that offered distinct temperature environments separated by fluid dynamics instead of physical barriers. (**B**) Dye flow test highlights eight distinct thermal zones created using concentric flow directed towards the centre of the apparatus. (**C**) Analysis of temperature stability for established thermal zones. Single lines correspond to highest (26°C) and lowest (16°C) temperatures. Double lines denote values from equivalent zones on opposing sides of the apparatus for 19, 21, and 23°C target temperatures. Raw data included as *Figure 1—source data 1*. (**D**) Representative images of a fish (blue arrows) in raw infrared and processed video, under simulated daylight and night (full dark) conditions. Yellow identifier denotes strong tracking signal achieved for experimental setup. Red dot denotes centre point used to set coordinates for raw behavioural data.

The online version of this article includes the following source data for figure 1:

**Source data 1.** Numerical data contributing to *Figure 1*.

animal model where changes during induction and resolution phases of acute inflammation could be examined (*Havixbeck et al., 2017*; *Havixbeck et al., 2016*).

Behavioural examination identified four distinct phases of the fever response among groups of fish challenged in vivo with *Aeromonas veronii* (*Figure 2A*). Thermal selection patterns across challenged fish were remarkably reproducible, even when fish were placed individually in the annular swim chamber (*Figure 2B*). A distinct period from 1 to 8 days post-infection (dpi) emerged, where *Aeromonas*-challenged fish displayed a 2–3°C increase in thermal preference compared to mock-infected (saline)

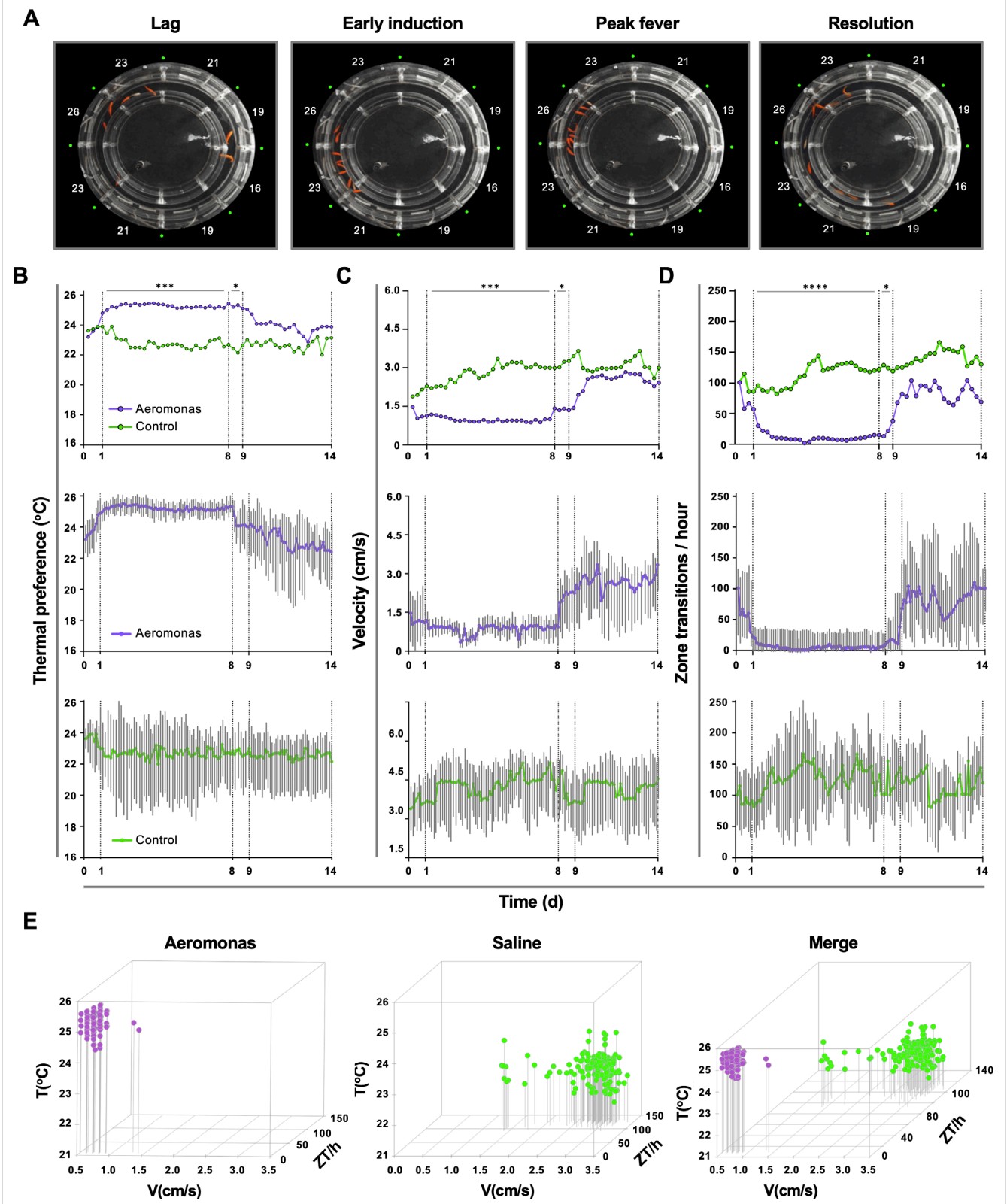

**Figure 2.** Homogeneity in thermal preference and sickness behaviours among fish eliciting fever. (**A**) Fish infected with *Aeromonas veronii* were free to select a range of environmental temperatures within the annular temperature preference tank (ATPT). Video still images show collective positioning of fish during distinct phases of fever response. Labels correspond to mean temperature for each ATPT thermal zone. (**B**) Temperature preference, (**C**) velocity, and (**D**) total transitions across thermal zones for fish infected with *Aeromonas* (*n* = 5) or mock infected with saline (*n* = 5). Fish were placed

*Figure 2 continued on next page*

*Figure 2 continued*

separately in the ATPT and individually monitored for 2 weeks. Evaluation of behavioural variance shows distinct periods of marked consistency in temperature preference, swimming velocity, and thermal zone transitions across *Aeromonas*-infected fish. Solid lines represent mean hourly values and vertical grey bars correspond to standard deviation at each time point. Results were analyzed by an ordinary two-way analysis of variance (ANOVA) and Šídák's multiple comparisons test. *p < 0.05, p < 0.01, ***p < 0.001, ****p < 0.0001. (**E**) Simultaneous three-parameter representation of behaviour hourly data points for *Aeromonas*-infected and saline control fish (*n* = 5 fish and 168 data points per group; mean values for 5 fish shown). 3D plots correspond to 1–8 dpi febrile period. Merged graph highlights segregation of behavioural responses. Figure produced using R (version 3.3). Raw data and open-source code included as *Figure 2—source data 1–5*.

The online version of this article includes the following source data for figure 2:

**Source data 1.** Numerical data contributing to *Figure 2*.

**Source data 2.** Numerical data contributing to *Figure 2*.

**Source data 3.** Numerical data contributing to *Figure 2*.

**Source data 4.** Numerical data contributing to *Figure 2*.

**Source data 5.** Code contributing to *Figure 2*.

controls (*Figure 2B*). Variance analysis confirmed the consistency in environmental temperature selected by individual animals within this 1–8 dpi time window (*Figure 2B*). Outside of this window, we found no significant difference in thermal selection between challenged and control individuals, with both *Aeromonas* and saline-treated groups also shifting to large fluctuations both temporally and among individual fish (*Figure 2B*).

Our behaviour analyses further identified two new measurable lethargy-associated outcomes in teleost fish, which add to the similarities between ectotherm and endotherm fever. The first was defined by a decrease in swimming velocity (V) among *Aeromonas*-challenged fish (1–8 dpi; *Figure 2C*). In contrast, velocity among control fish remained high and variable across individuals during the same timeframe (*Figure 2C*). The second lethargy parameter was based on changes to temperature seeking behaviour, defined by the rate of transitions that a fish made between distinct ATPT thermal zones. Whereas control saline injected fish continued to show one hundred or more zone transitions (ZT) per hour, *Aeromonas*-treated fish displayed a dramatic decrease in the number of ZT during the same 1–8 dpi window (*Figure 2D*). As with temperature preference and velocity measurements, ZT values within this fever behavioural window were remarkably consistent across individual *Aeromonas*-challenged fish, increasing in variance after 8 dpi (*Figure 2D*). In sharp contrast, control fish displayed significant heterogeneity throughout the entire observation period. These two new lethargy-associated metrics are consistent with established sickness behaviours of metabolic fever in humans and other endotherms (immobility, fatigue, and malaise) (*Harden et al., 2015*) and help to further define the behavioural fever response of teleost fish.

Hourly values for *Aeromonas*-challenged and saline control groups were evaluated simultaneously during the established fever window (1–8 dpi) and across the broader 14-day observation period. Between 1 and 8 dpi, we identified marked segregation in the responses elicited by fish in these two groups (*Figure 2E*). For *Aeromonas*-challenged fish, V and ZT values remained exclusively low during the febrile period (*Figure 2E*). In contrast, saline control fish exhibited a wider range of movement profiles, dominated by high V and ZT values (*Figure 2E*).

## Activation of CNS and systemic febrile programmes following *Aeromonas* infection

To confirm classic engagement of the CNS via fever and assess potential differences with mechanical FRH, hypothalamic tissue was isolated from *Aeromonas*-challenged fish and examined for local expression of pyrogenic cytokines. The selected genes, *il1b*, *tnfa*, and *il6*, showed more robust local induction of gene expression under dynamic fever conditions ($T_D$ group), where fish had been allowed to swim freely through the established 10°C temperature gradient within the ATPT (*Figure 3A*). Two cytoprotective elements (*hsp70* and *hsp90*) further displayed the highest levels of expression in the hypothalamus under these dynamic thermal conditions. Evaluation of $PGE_2$ concentrations in systemic circulation showed an early peak at 24 hr post-infection (hpi) for $T_D$ fish, consistent with its role as a major pyrogenic mediator of fever (*Evans et al., 2015*; *Figure 3B*). These responses were distinct from those fish placed under 26°C ($T_{S26}$; mechanical FRH) or 16°C static thermal conditions ($T_{S16}$; basal

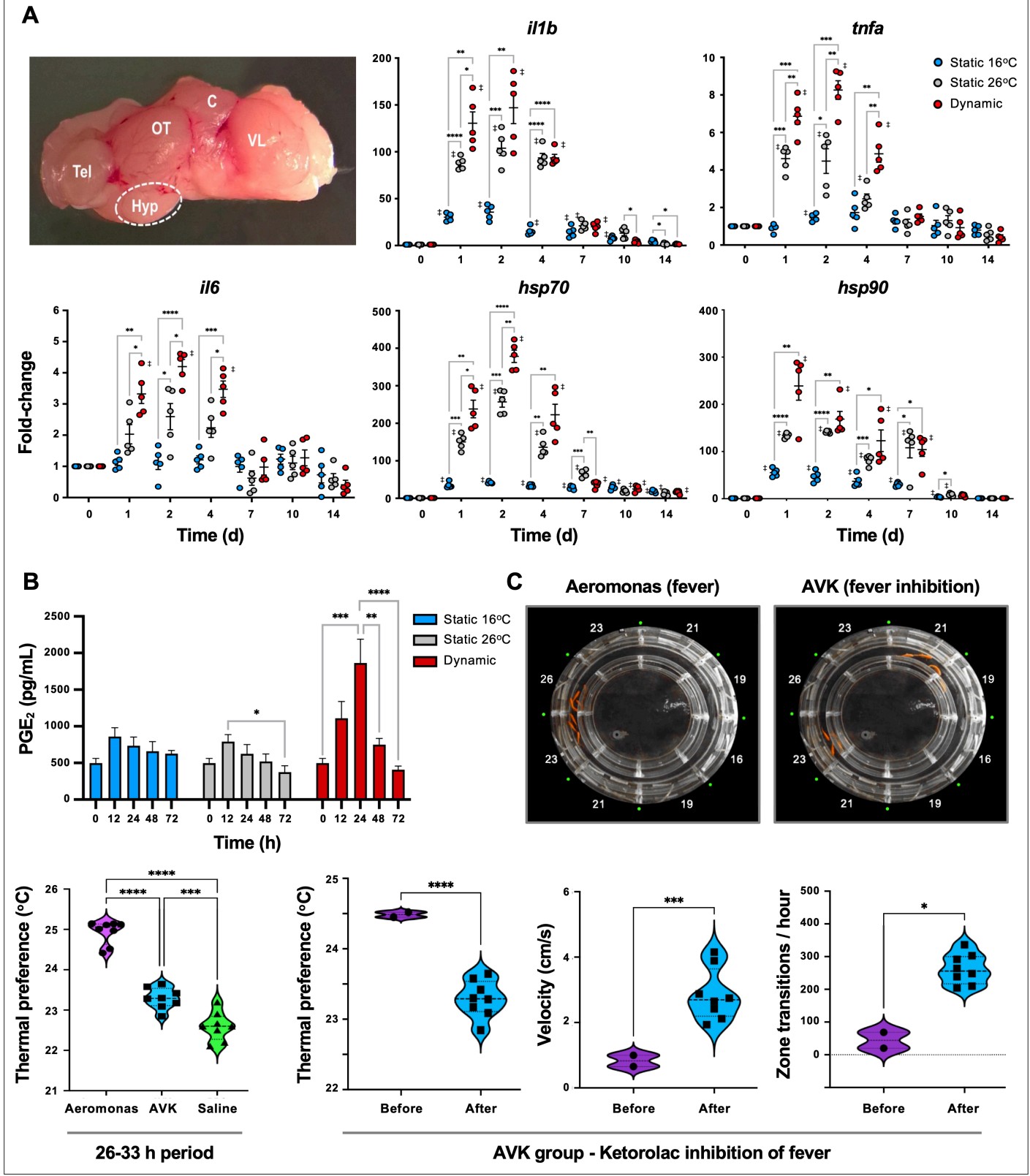

**Figure 3.** Confirmation that fish fever engages central nervous system (CNS) and systemic pyrogenic signals following *Aeromonas* infection. Fish were inoculated and placed in static 16°C (basal acclimated temperature), static 26°C (mechanical hyperthermia; maximum temp. that fish selected during behavioural fever), or dynamic fever (where fish could move freely between thermal zones). (**A**) Quantitative PCR (qPCR) evaluation of hypothalamic responses following infection (*n* = 5 per group per time point; 3 technical replicates per fish per time point). Symbols correspond to individual samples;

*Figure 3 continued on next page*

*Figure 3 continued*

lines represent the mean ± standard error of the mean (SEM); *actinb* served as reference gene. Results analyzed by an ordinary two-way analysis of variance (ANOVA) using Tukey post hoc test. (**B**) Plasma collected at 0, 12, 24, 48 and 72 hr post-infection (*n* = 6 per group per time point) and prostaglandin E2 (PGE$_2$) concentrations determined via ELISA. Results analyzed by an ordinary one-way ANOVA followed by a Tukey's post hoc test. Error bars represent + SEM. (**C**) Confirmation that an antipyretic inhibits fish fever. Video still images illustrate disruption of high thermal preference collective positioning following treatment of *Aeromonas*-infected fish with ketorolac tromethamine. Quantitation of thermal preference shows differences between fish infected with *Aeromonas*, those infected with *Aeromonas* and treated with ketorolac (AVK), and those mock infected with saline (*n* = 5 fish per group per time point). Observation period began 2 hr after start of peak febrile response (26 hr) and continued for 8-hr duration of ketorolac action. One way ANOVA followed by Tukey's multiple comparisons test compared *Aeromonas*, AVK, and saline groups. Changes were further evaluated within AVK group before and after ketorolac administration. Welch's *t*-tests compared behavioural parameters. For all, *p < 0.05, **p < 0.01, ***p < 0.001, ****p < 0.0001; ‡ denotes significant difference from time 0, p < 0.05. Raw data included as *Figure 3—source data 1–3*.

The online version of this article includes the following source data and figure supplement(s) for figure 3:

**Source data 1.** Numerical data contributing to *Figure 3*.

**Source data 2.** Numerical data contributing to *Figure 3*.

**Source data 3.** Numerical data contributing to *Figure 3*.

**Figure supplement 1.** Thermal increases alone are not sufficient to engage central nervous system (CNS) and stimulate immune regulatory and cytoprotective genes.

**Figure supplement 1—source data 1.** Numerical data contributing to *Figure 3—figure supplement 1*.

acclimated temperature) following infection. T$_{S26}$ FRH promoted upregulation of cytokine and cytoprotective genes in our panel, but to lower levels than those fish allowed to exert dynamic fever (*Figure 3A*). Circulating PGE$_2$ concentrations remained near basal levels for both T$_{S26}$ FRH and T$_{S16}$ groups (*Figure 3B*). Hyperthermia alone without pathogen stimulus was not sufficient to activate these cytokines and cytoprotective genes in the CNS (*Figure 3—figure supplement 1*).

Administration of an antipyretic offered added support for the shared biochemical pathways directing fever in ectotherms and endotherms. We chose ketorolac as an NSAID with the capacity to inhibit COX-1 and COX-2 (*Vadivelu et al., 2015*). This drug has been successfully used in a range of animal species (*Rooks, 1990*) and can be injected (*Vadivelu et al., 2015*; *Baevsky et al., 2004*), thereby ensuring consistent dosing. In humans, a dose of 0.5 mg/kg is effective, and results in a 15- to 20-min onset with a 6- to 8-hr duration of action (*Vadivelu et al., 2015*). Similarly, injection of ketorolac to *Aeromonas*-infected fish inhibited fever at a dose of 0.5 mg/kg (Aeromonas vs. AVK experimental group; *Figure 3C*). Examination of changes within the AVK group before and after ketorolac administration further showed that this NSAID inhibited fever-associated increases in thermal preference and lethargy behaviours (*Figure 3C*).

## Fever markedly improves *Aeromonas* clearance while showing selectivity in the induction of reactive oxygen species and nitric oxide antimicrobial defenses

During the course of infection, furuncles caused by *Aeromonas* species can shed up to 10$^7$ bacteria per hour in fish (*Rose et al., 1989*). Thus, we assessed the presence of *A. veronii* on the furuncle surface as a measure of pathogen load and shedding potential. Infected fish held at 16°C static thermal conditions (T$_{S16}$) displayed heavy bacterial loads through the first 4 days after infection, reducing to 70 ± 23 and 28 ± 10 CFU at 7 and 10 dpi, respectively, and subsequently progressing below detectable levels by 14 dpi (*Figure 4A*). Fish in the dynamic fever group (T$_D$) also showed heavy initial bacterial burden, but these decreased markedly faster compared to T$_{S16}$ fish (40 ± 27 CFU at 4 dpi, and below detectable levels by 7 dpi). Mechanical hyperthermia (T$_{S26}$) yielded an intermediate response, achieving *Aeromonas* clearance 10 days following infection (*Figure 4A*). Thus, fish allowed to exert dynamic fever cleared *A. veronii* in half the time than fish maintained in static 16°C conditions. Notably, this enhancement in clearance could not be explained by the current thermal restriction model since *A. veronii* showed faster growth as incubation temperatures increased from 16 to 26°C (*Figure 4B*).

To assess the contributions of fever to immune cell function, we first quantitated their recruitment into the mucosal infection site. Both T$_D$ and T$_{S26}$ fish showed equivalent accelerated infiltration kinetics compared to *Aeromonas*-infected fish held at 16°C static thermal conditions (*Figure 4C* and *Figure 4—figure supplement 1*). To determine if this enhancement was paired with activation of

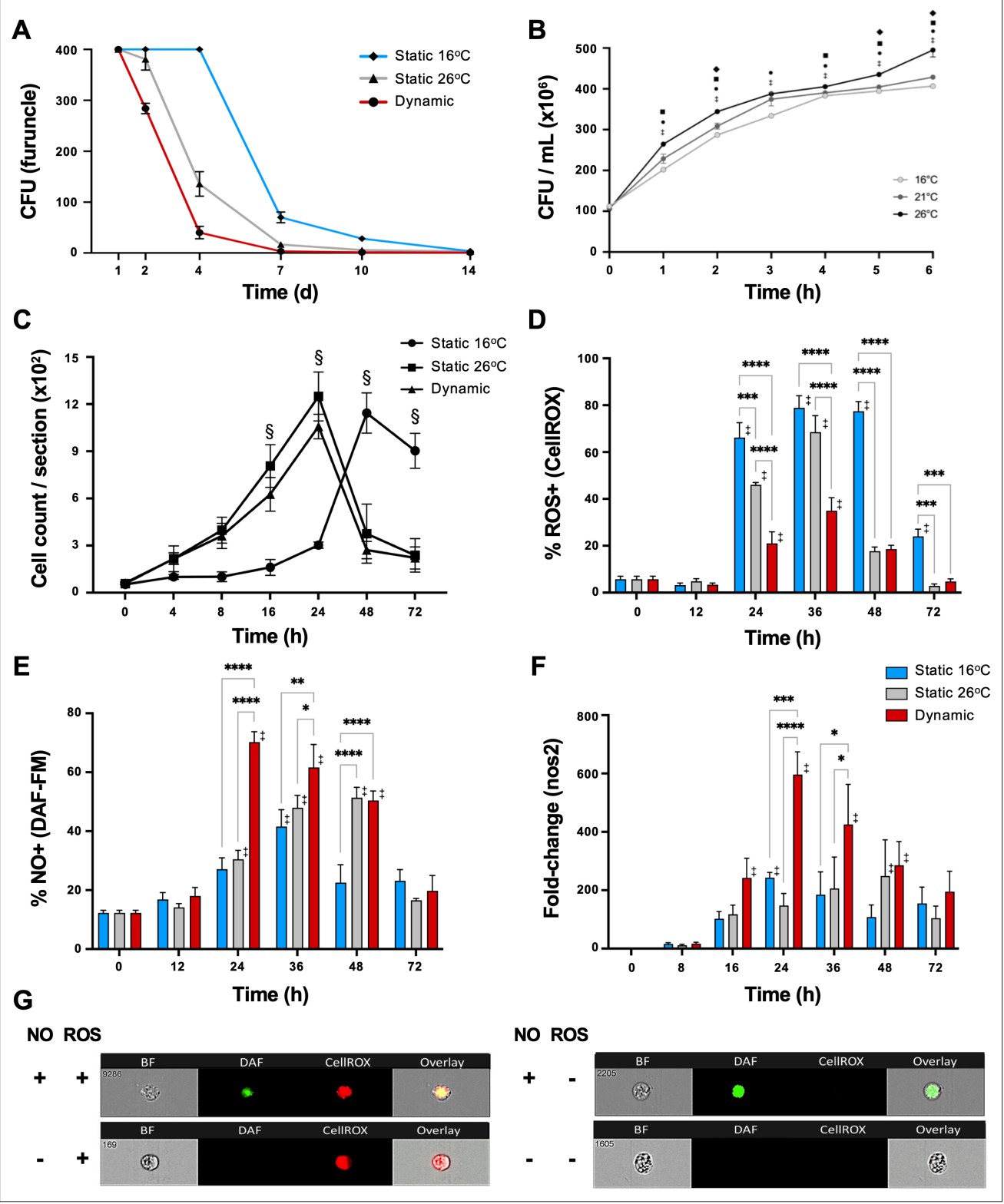

**Figure 4.** Fever enhances pathogen clearance while showing selectivity in the induction of reactive oxygen species (ROS) and nitric oxide (NO) antimicrobial defenses. Fish were infected with *Aeromonas* and placed under 16°C static, 26°C static (mechanical hyperthermia), or dynamic host-driven fever conditions. (**A**) Bacterial loads and pathogen shedding potential assessed following surface sampling of infection site. Mean values ± standard error of the mean (SEM) shown (*n* = 5 per group per time point; 3 technical replicates per fish per time point). (**B**) Effect of temperature on *Aeromonas* growth assessed in vitro (*n* = 3 per group per time point). Results analyzed by an ordinary two-way analysis of variance (ANOVA) using Tukey post hoc

*Figure 4 continued on next page*

*Figure 4 continued*

test. Differences correspond to statistical significance of $p < 0.05$ between 16 vs. 26°C (•), 16 vs. 21°C (♦), and 21°C vs. 26°C (■); ‡ denotes significant difference from time 0, $p < 0.05$. (**C**) Hematoxylin and eosin were used to stain infected wounds. ImageJ analysis assessed differential cell recruitment ($n = 3$ per group per time point); refer to *Figure 4—figure supplement 1* for source data. Results analyzed by an ordinary two-way ANOVA using Tukey post hoc test. § corresponds to statistical significance of $p < 0.01$ when dynamic and static 26°C groups were compared to 16°C static conditions. No significant differences found between dynamic and static 26°C groups at any time point. Following *Aeromonas* challenge, cutaneous leukocytes were isolated from fish held at 16°C static, 26°C static, or fever dynamic thermal conditions. Imaging flow cytometry evaluated leukocyte production of (**D**) ROS via CellROX, and (**E**) nitric oxide production via DAF-FM-DA ($n = 5$ per group per time point). (**F**) qPCR analysis of wound tissue shows kinetics gene encoding inducible nitric oxide synthase (iNOS; $n = 5$ per group per time point; 3 technical replicates per fish per time point; *actinb* served as reference gene). (**G**) ImageStream MKII flow cytometer images show positive and negative cells after CellROX and DAF-FM-DA staining. Results analyzed using a two-way ANOVA followed by Tukey's post hoc test. *$p < 0.05$, **$p < 0.01$, ***$p < 0.001$, ****$p < 0.0001$; ‡ denotes significant difference from time 0, $p < 0.05$. Raw data included as *Figure 4—source data 1–4*.

The online version of this article includes the following source data and figure supplement(s) for figure 4:

**Source data 1.** Numerical data contributing to *Figure 4*.

**Source data 2.** Numerical data contributing to *Figure 4*.

**Source data 3.** Numerical data contributing to *Figure 4*.

**Source data 4.** Numerical data contributing to *Figure 4*.

**Figure supplement 1.** Thermal promotion of leukocyte recruitment to skin wounds in *Aeromonas*-infected fish.

antimicrobial pathogen killing mechanisms, we then examined the production of reactive oxygen species (ROS) by infiltrating leukocytes, as a prominent, effective, and evolutionarily conserved innate defense mechanism (*Fang, 2004*; *Neumann et al., 2001*). As previously shown by us and others (*Havixbeck et al., 2017*; *Havixbeck et al., 2016*; *Neumann et al., 2001*), leukocytes derived from fish housed under thermally static conditions ($T_{S16}$ fish for this study) display strong capacity for generation of ROS; over 75% of peritoneal leukocytes were positive for ROS production in the *A. veronii* in vivo cutaneous challenge model (*Figure 4D*). Mechanical hyperthermia ($T_{S26}$) yielded a similar response, with prominent ROS production peaking at 36 hpi (*Figure 4D*). Surprisingly, the number and proportion of ROS-producing leukocytes were greatly reduced under host-driven dynamic thermoregulatory conditions ($T_D$; *Figure 4D*) despite the enhanced kinetics in leukocyte recruitment outlined above (*Figure 4C*).

Given the long-established contributions of fever to host survival (*Evans et al., 2015*; *Kluger et al., 1975*; *Earn et al., 2014*), we hypothesized that fever may promote an innate antimicrobial response that does not include a strong ROS production component. Thus, we also evaluated leukocyte nitric oxide (NO) production, as an alternative evolutionarily conserved innate response to pathogen attack (*Neumann et al., 2001*; *Bogdan, 2015*). Contrary to results for ROS, we identified greater overall levels as well as accelerated kinetics of NO production under fever conditions (*Figure 4E*). Leukocytes infiltrating the furuncle of $T_D$ fish showed significant upregulation, with peak NO production observed at 24 hpi (*Figure 4E*). This was further supported by a marked, earlier upregulation of the gene encoding inducible nitric oxide synthase (iNOS; *nos2*), which is necessary for production of immune NO (*Figure 4F*; *Neumann et al., 2001*; *Bogdan, 2015*). In sharp contrast, both $T_{S16}$ and $T_{S26}$ fish displayed lower levels of *nos2* expression and lower overall capacity to produce NO (*Figure 4E–G*). Thus, fever differentially regulated ROS and NO leukocyte antimicrobial mechanisms in *Aeromonas*-challenged fish.

## Fever promotes earlier resolution of acute inflammation

To date, studies looking at the basis for host survival due to fever have focussed on the activation of immune defense mechanisms. The self-resolving nature of our teleost model allowed us to also characterize immunological changes during the transition between induction and resolution phases of acute inflammation. Indeed, comparison of cellular responses following *Aeromonas* infection showed differences in the control of leukocyte recruitment to the infection site. $T_D$ and $T_{S26}$ fish reached the peak of infiltration at 24 hpi, subsequently decreasing and nearing basal levels by 48 hpi (*Figure 4C*). This was consistent with faster kinetics of induction and control of local *tnfa*, *il1b*, and *cxcl8* gene expression among these fish (*Figure 5A*). In contrast, $T_{S16}$ fish showed slower kinetics of leukocyte recruitment with a delayed peak at 48 hpi that was further sustained beyond 72 hpi (*Figure 4C*). These fish also showed delayed upregulation and control of local *tnfa*, *il1b*, and *cxcl8* gene expression; the latter two

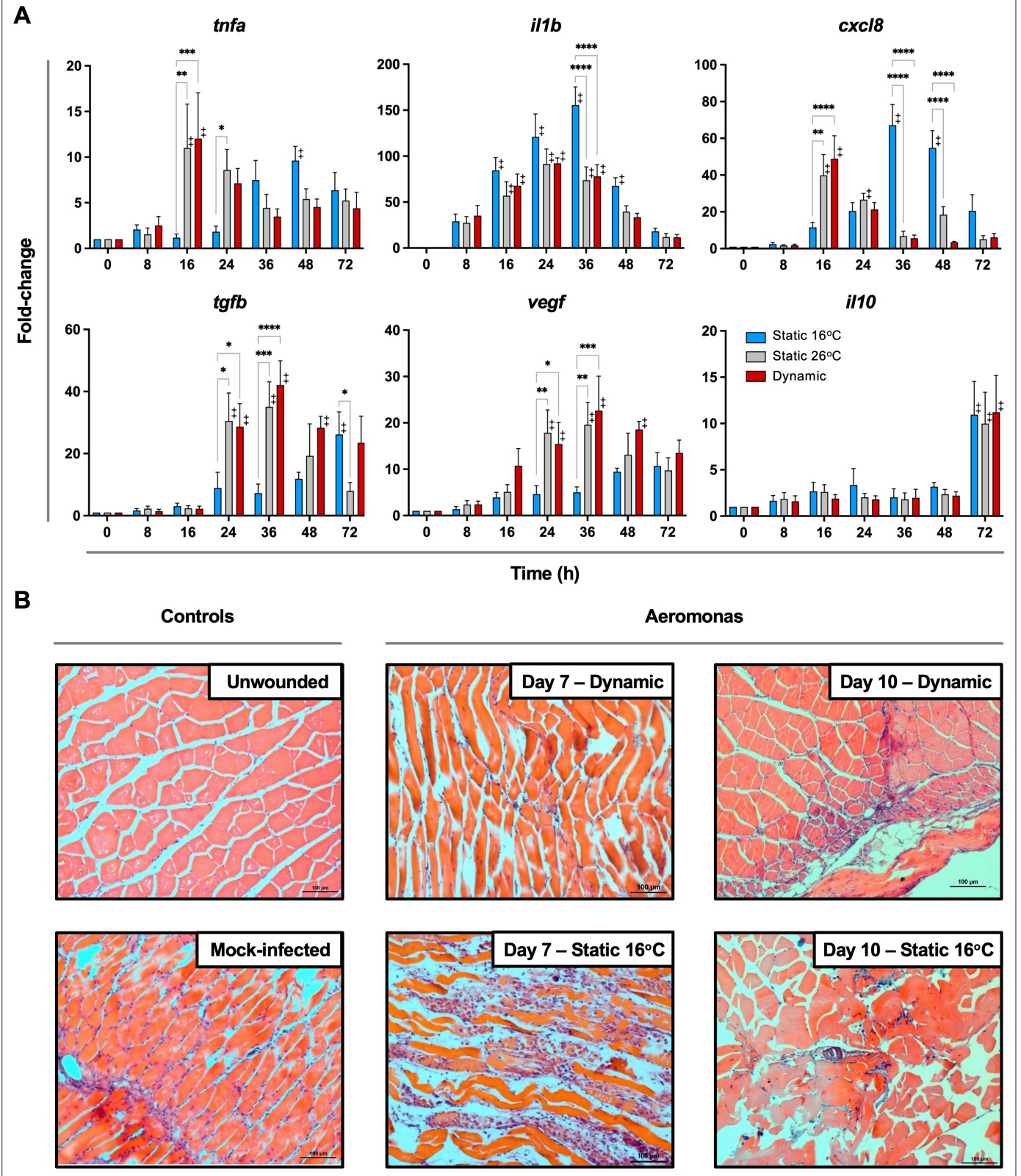

**Figure 5.** Fever promotes inflammation control and shows early engagement of tissue repair regulators following *Aeromonas* infection. (**A**) Cutaneous wounds were evaluated for expression of genes encoding pyrogenic cytokines tumor necrosis factor alpha (TNFA) and interleukin-1 beta (IL1B), CXCL8 chemokine, and pro-resolution cytokines VEGF, TGFB, and IL10 via qPCR ($n = 5$ per group per time point; 3 technical replicates per fish per time point; *actinb* served as reference gene). Data were analyzed with an ordinary two-way analysis of variance (ANOVA) using a Tukey post hoc test. *$p < 0.05$,

*Figure 5 continued on next page*

*Figure 5 continued*

\*\*p < 0.01, \*\*\*p < 0.001, \*\*\*\*p < 0.0001; ‡ denotes significant difference from time 0, p < 0.05. Raw data included as *Figure 5—source data 1*. (**B**) Histopathological assessment highlights inflammation, skin barrier damage, and repair at mid and late stages of the infection process. Hematoxylin and eosin (H&E)-stained tissues sectioned from fish inoculated with *Aeromonas* and allowed to exert fever (dynamic) or placed under 16°C static conditions. H&E-stained tissues sectioned from healthy controls (day 0; non-inoculated) or mock-infected fish (day 0 saline) are provided as controls. *n* = 3 for each group. Scale bar: 100 μm.

The online version of this article includes the following source data for figure 5:

**Source data 1.** Numerical data contributing to *Figure 5*.

pro-inflammatory cytokines further displayed markedly higher levels of expression (*Figure 5A*). Finally, we identified earlier and more pronounced expression of the anti-inflammatory cytokine (*tgfb*) and the pro-reparative vascular endothelial growth factor (*vegf*) among $T_D$ and $T_{S26}$ fish (*Figure 5A*).

Histological examination of experimental wounds further supported marked differences in the inflammation control process of fish maintained in static ($T_{S16}$) and dynamic ($T_D$) temperature conditions (*Figure 5B*). Whereas unwounded controls had normal muscle fibres and no signs of inflammation, large infiltrates of granulocytes and macrophages remained in the subdermal muscle tissue of $T_{S16}$ fish 7 days after cutaneous infection with *Aeromonas* (*Figure 5B*). Groups of inflammatory infiltrates were evident in the extracellular space between muscle fibres as well as over damaged muscle fibres. This was in stark contrast to fish allowed to exert fever. Day 7 wounds from these $T_D$ fish showed less tissue damage and only traces of the original leukocyte infiltrates remained (*Figure 5B*), thereby resembling those found in the uninfected wounded day 0 control fish (*Figure 5B*). By day 10, infected wounds from fish housed under 16°C static thermal conditions showed prominent necrosis among muscle fibres, edema, and some immune cell infiltrates (*Figure 5B*). In contrast, wounds from

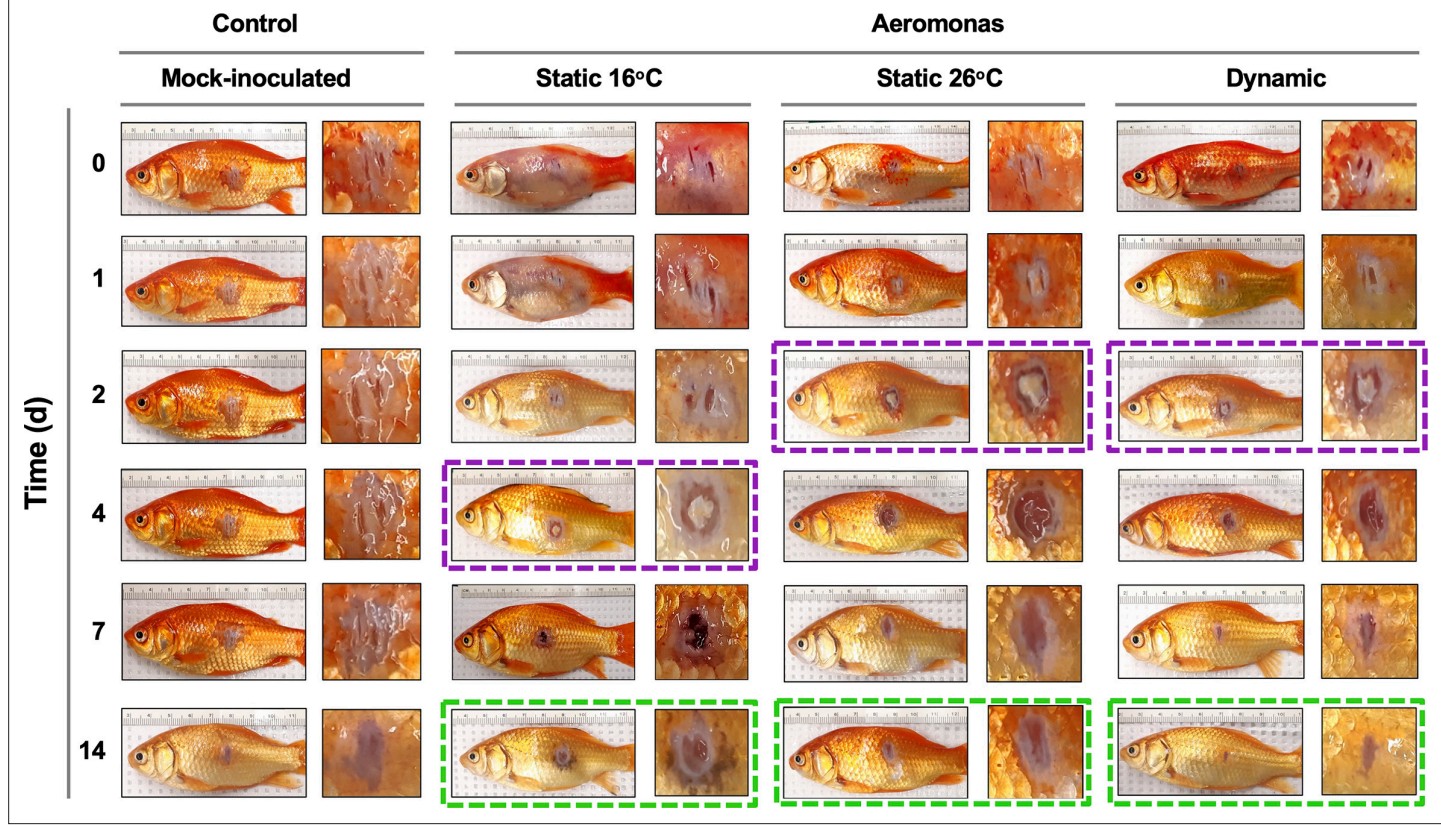

**Figure 6.** Progression of tissue pathology in *Aeromonas*-infected fish. Representative images show focal gross lesions for fish inoculated with *Aeromonas veronii* and placed under 16°C static, 26°C static (mechanical fever-range hyperthermia), or dynamic host-driven fever conditions. Fish mock infected with saline are provided as controls. Time points capture progression from initial infection to advanced stages of wound repair. Purple boxes highlight differential kinetics of purulent exudate formation. Green boxes showcase distinct degrees of wound closure achieved across *Aeromonas*-infected groups by 14 dpi.

fish allowed to exert fever had no necrotic regions and inflammation had largely resolved (**Figure 5B**). Thus, fever promoted an earlier pro-inflammatory period, which was further paired with more efficient resolution of inflammation based on inhibition of leukocyte recruitment, control of pro-inflammatory cytokine expression, induction of pro-resolution genes, and management of collateral tissue damage.

## Fever enhances wound repair

Characterization of pathology at the infection site showed marked differences in the capacity of $T_D$ fish to heal *A. veronii*-associated wounds. Similar levels of inflammation were evident in furuncles of $T_D$, $T_{S26}$, and $T_{S16}$ fish 1 day after cutaneous infection (**Figure 6**). However, consistent with the enhanced kinetics of leukocyte recruitment (**Figure 4C**), $T_D$ and $T_{S26}$ fish showed accelerated kinetics of purulent exudate formation by 2 dpi. Fish exerting dynamic fever subsequently progressed most rapidly, displaying early signs of tissue repair and scale regeneration by 7 dpi, and advanced stages of wound healing by 14 dpi (**Figure 6**). Comparatively, $T_{S26}$ and $T_{S16}$ furuncles did not reach equivalent stages of wound closure (**Figure 6**; green boxes).

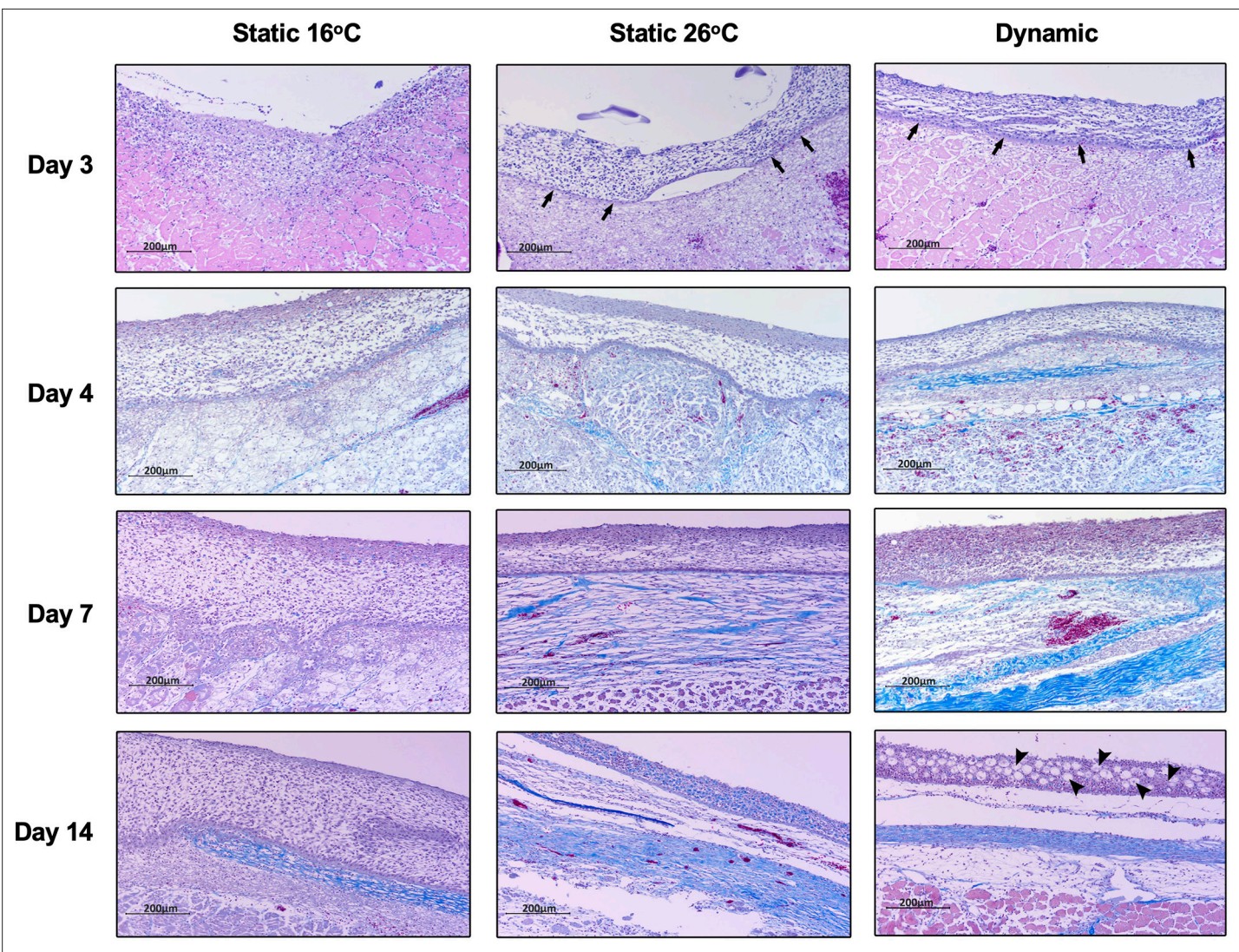

**Figure 7.** Fever enhances re-epithelialization and collagen deposition. Wound tissue from *Aeromonas*-infected fish was collected at the indicated time points, sectioned, and stained with Masson's Trichrome stain (*n* = 3 per group per time point). Histopathological examination highlights early development of basal layer of epidermis (arrows; day 3) and overlying layers of keratinocytes among fish in 26°C static and dynamic host-driven fever conditions. Subsequent differential progression of collagen deposition is shown by the increased abundance and organization of blue-stained fibres. Restitution of mucus producing cells in epidermis is highlighted by arrowheads (day 14). Scale bar: 200 μm.

Thus, fish allowed to exert fever resolved *Aeromonas* infection and repaired the associated skin barrier damage faster than those maintained under mechanical FRH or static 16°C temperature housing conditions.

Histopathological examination of *Aeromonas*-infected furuncles stained using Masson's Trichrome stain demonstrated earlier re-establishment of the basal epidermal layer and overlying layers of keratinocytes among $T_D$ and $T_{S26}$ experimental groups (day 3; *Figure 7*). Among $T_{16}$ fish, wound re-epithelialization occurred but was delayed. As per surface pathology described above, tissue repair subsequently advanced faster in wounds derived from $T_D$ fish, with de novo collagen synthesis becoming evident as early as 4 dpi (*Figure 7*). This further developed into more extensive, organized collagen deposition by 7 dpi (*Figure 7*). In comparison, slower progression was observed among $T_{S26}$ and $T_{16}$ wounds based on the abundance and relative organization of collagen arrangements at the wound site. Regeneration of mucus-secreting cells, consistent with re-establishment of skin barrier functionality, was only observed in the $T_D$ experimental group (day 14; *Figure 7*). Thus, fever promoted greater levels of wound repair and regained original structural features required for restoration of skin barrier functionality after cutaneous infection. Conversely, the absence of fever caused delayed resolution of the inflammatory response, re-epithelialization, and the appearance of extracellular matrix components. Together, our results show that in this cold-blooded vertebrate moderate self-resolving fever offers a natural strategy to harness the body's intrinsic repair mechanisms to enhance wound healing.

## Discussion

Induction-centric models cannot explain the continued selection of fever through 550 million years of metazoan evolution, given the predicted high energetic costs of a hypermetabolic state and the implications of inflammation-associated tissue damage (*Wang and Medzhitov, 2019*; *Kluger, 1979*; *Muchlinski, 1985*). A recent emphasis on pathological forms of fever (highest temperatures or sustained hyperthermia; severe disease phenotypes) (*Bone, 1996*; *Liu et al., 2012*; *Singer et al., 2016*) further shift our focus away from the more common moderate forms of this natural biological process (*Evans et al., 2015*; *Hasday et al., 2014*; *Islam et al., 2021*). Thus, for this study we examined fever under an acute inflammatory state that was transient and self-limiting. A eurythermic teleost fish model offered fine control of febrile mechanisms through host-driven dynamic thermoregulation, more closely mimicking natural conditions for heating and cooling. A custom animal enclosure delivered multi-day thermal gradient stability without the use of physical barriers that are known to affect behaviour (*Rakus et al., 2017*; *Myrick et al., 2004*). Together, this combination of animal model, enclosure design, and automated continuous per second tracking of animal locomotory patterns over both induction and resolution phases of the febrile period greatly enhanced analytical robustness and temporal resolution compared to previous studies. Collectively, our data show that moderate self-resolving fever offers earlier and selective rather than stronger induction of innate antimicrobial programmes against infection, and that this results in markedly faster pathogen clearance. We also show that this is paired with rapid subsequent inflammation control and enhanced tissue repair at the wound site. This integrative approach is novel and represents a marked upgrade in refinement compared to popular hypotheses that postulate global induction of immunity, or rely on a shift away from those temperatures preferred by invading pathogens. Our demonstration that this integrative strategy was already well established prior to the development of endothermy further suggests a plausible scenario for the long-standing selection of fever through evolution.

Increases in core body temperature are well established to promote neutrophil accumulation, NADPH oxidase activity, and production rates of toxic superoxide anions (*Souabni et al., 2017*). Models of FRH have linked these effects to higher serum concentrations of IL-1, TNF, and IL-6 (*Jiang et al., 2000*; *Ostberg and Repasky, 2000*), granulocyte colony-stimulating factor (G-CSF)-driven release of neutrophils from the hematopoietic bone marrow, *Capitano et al., 2012*; *Ellis et al., 2005* expansion of the circulating neutrophil pool (*Capitano et al., 2012*; *Ellis et al., 2005*), increased endothelial barrier permeability in blood vessels (*Shah et al., 2012*), and upregulation of granulocyte-macrophage colony-stimulating factor (GM-CSF), extracellular HSP70, IL-8, and other CXC chemokines at the local site of infection (*Hasday et al., 2003*; *Lee et al., 2012*). However, these thermal increases have also been shown to promote collateral tissue injury (*Hasday et al., 2014*; *Lipke et al., 2010*; *Rice et al., 2005*). This continues to be broadly viewed as an unavoidable cost of fever, where

induction of immune defenses will undoubtedly drive accompanying inflammation-associated tissue injury (*Hasday et al., 2003*; *Rice et al., 2005*). Our results offer an alternative explanation where, in the absence of natural thermoregulation, FRH only partially recapitulates the regulatory capacity of fever on acute inflammation. We found that both fever and mechanical FRH displayed accelerated kinetics of leukocyte recruitment, and earlier control of pro-inflammatory cytokine expression. But differences were observed in the induction of pro-resolution genes, and fever ultimately achieved greater levels of wound repair. We also documented a clear shift from an ROS-dominant microbicidal response under euthermic and FRH conditions to one dominated by NO production under fever. NO and downstream reactive nitrogen species exert microbicidal or microbiostatic activity against a broad range of bacteria, viruses, yeasts, helminths, and protozoa (*Wink et al., 2011*). Yet, at first glance, this inhibition in ROS production seemed contradictory to the role of fever in pathogen resistance. Notably, decreased ROS activity does not necessarily have to result in compromised host defenses. In a murine *Pseudomonas aeruginosa* infection, NO enhanced bacterial clearance via an Atg7-mediated mechanism that also reduced IFN-γ activity,inhibited ROS production, and limited oxidative stress, resulting in decreased lung injury and lower infection-associated mortality (*Li et al., 2015*). A growing number of examples for antagonistic NO modulation of ROS responses during infection have also been described, some of which can be traced as far back as plants (*Fang, 2004*; *Wink et al., 2011*; *Clancy et al., 1998*; *Graham et al., 2018*; *Kolbert et al., 2019*). Our results are consistent with these observations and offer a natural scenario where fever drives a shift in the NO–ROS balance that maintains competencies in microbial clearance to subvert a live infection, while also contributing to inflammation control and the re-establishment of a functional mucosal barrier.

Recent years have seen renewed interest in the links between thermoregulation and host defense, which now also extend to cooler temperatures (*Steiner and Romanovsky, 2019*; *Wang and Medzhitov, 2019*; *Liu et al., 2012*; *Ganeshan et al., 2019*; *Medzhitov et al., 2012*). This has led to a perceived functional dichotomy between fever and hypothermia. Although both reflect an animal's capacity to take advantage of thermoregulation to maintain fitness upon infection, fever drives elimination of invading microorganisms through microbicidal disease resistance while hypothermia promotes tolerance to foster energy conservation and management of collateral inflammation-associated tissue damage (*Steiner and Romanovsky, 2019*; *Wang and Medzhitov, 2019*). Our results blur the line in this perceived dichotomy. We identified clear fever-embedded mechanisms that contributed to the maintenance of tissue integrity and would allow for energy conservation. These contributions did not appear in response to damage, but instead were prominent features through initial induction and subsequent resolution phases of acute inflammation. Early on, fever promoted disease resistance via enhanced kinetics of leukocyte recruitment rather than by augmenting the magnitude or duration of immune activation. Engagement of cytoprotective gene programmes in the CNS occurred within hours of the initial immune challenge. Selective rather than global upregulation of immunity further decreased the potential for collateral damage often attributed to fever. Notably, microbicidal efficacy was still superior under fever, with *Aeromonas* clearance achieved markedly faster than under thermally restricted basal conditions (7 vs. 14 days). Our results then showed greater efficiency in inflammation control. This was paired with novel contributions that actively promoted tissue repair rather than resilience to inflammatory damage. There was no induction of a hypothermic state at any point during our observation periods; instead, high-resolution positional tracking only showed a discrete, self-resolving fever response. Thus, fever actively engages mechanisms that enhance protection, while also limiting pathology, controlling inflammation, and promoting tissue repair. Importantly, our findings do not dispute previously described competition mechanisms between immunity and other maintenance programmes that direct a transition towards tolerance under high pathogen loads (*Liu et al., 2012*; *Romanovsky and Székely, 1998*). Instead, they simply highlight the persistent and long-standing considerations placed on energy allocation and tissue integrity by an animal host at all stages of infection.

To conclude, our results reveal novel features of fever, and demonstrate that it is an integrative host response to infection that regulates both induction and resolution phases of acute inflammation. Much work remains to establish the extent by which immune modulation through fever is conserved across cold- and warm-blooded vertebrates, despite their different physiologies. However, the downstream implications of our findings are potentially far-reaching. Among others, enhanced re-establishment of barrier integrity at tissue sites such as the skin are predicted to reduce the potential for secondary

infections, and curtail the physiological stress stemming from a need for extended management of wounded tissue. There are also inferences at the population level, where marked enhancements observed in pathogen clearance are likely to translate into lower rates of transmission across a naive population, offering novel opportunities for the management of infectious disease. More studies will be required to test these possibilities, but this is critical as we look to better understand the long-standing role of fever in the modulation of immunity, and the repercussions of inhibiting moderate fever in veterinary and human medicine.

# Materials and methods

**Key resources table**

| Reagent type (species) or resource | Designation | Source or reference | Identifiers | Additional information |
|---|---|---|---|---|
| Genetic reagent (Goldfish; *Carassius auratus*) | WT | Aquatic Imports | | 10–15 cm in length; mix-sex |
| Strain, strain background (*Aeromonas veronii*) | WT | Field isolated | NCBI Taxonomy ID: 114517 | |
| Chemical compound, drug | Ketorolac tromethamine | ATNAHS Pharma | Cat #: 2162644 | 0.5 mg/kg of body weight |
| Chemical compound, drug | Trizol | Thermo Fisher Scientific | Cat #: 15596026 | |
| Chemical compound, drug | CellROX Deep Red Reagent | Thermo Fisher Scientific | Cat #: C10491 | |
| Chemical compound, drug | DAF-FM DA | Invitrogen | Cat #: D23844 | |
| Commercial assay, kit | iScript cDNA synthesis kit | Bio-Rad | Cat #: 1708891 | |
| Commercial assay, kit | Prostaglandin E2 ELISA Kit | Cayman Chemical | Cat #: 514010 | |
| Software, algorithm | IDEAS Image Data Exploration and Analysis Software | IDEAS (https://www.luminexcorp.com/imagestreamx-mk-ii/#software) | RRID:SCR_019195 | |
| Software, algorithm | Ethovision XT software | Ethovision XT (https://www.noldus.com/ethovision) | RRID:SCR_000441 | |
| Software, algorithm | R software | R (http://www.r-project.org/) | RRID:SCR_001905 | |
| Software, algorithm | ImageJ software | ImageJ (http://imagej.nih.gov/ij/) | RRID:SCR_003070 | |
| Software, algorithm | GraphPad Prism software | GraphPad Prism (https://graphpad.com) | RRID:SCR_002798 | |

## Animals

Goldfish (*Carassius auratus auratus*), 10–15 cm in length, mix-sex, were purchased from Mt. Parnell Fisheries (Mercersburg, PA) and imported to Canada via Aquatic Imports (Calgary, Canada). They were held in a flow through system with simulated natural photoperiod (12 hr of light: 12 hr of dark) in the Aquatics Facility, Department of Biological Sciences, University of Alberta. Water quality parameters were sustained at 5.5–6.5 PPM dissolved oxygen and pH 7.2–8.0. Fish were fed with floating pellets once daily.

## Design and validation of the annular temperature preference apparatus

The ATPT was constructed from customized precision cut acrylic sheets molded and sealed into three concentric rings: an outer-most inflow ring separated into eight equal segments around the periphery, a middle continuous swim chamber containing no physical barriers, and an inner circle used to control depth. An additional inner compartment contained outflow and drainage. Small equidistantly drilled pores placed high on the outer inflow chambers and low on the inner outflow chamber allowed for water flow from the periphery to the centre of the apparatus. This ring-shaped swim chamber maintained constant water depth, current and perceived cover throughout, which are factors known to impact behaviour of aquatic animals. Eight distinct thermal zones were maintained by fluid dynamics. Temperatures of these zones were monitored on a per second basis over 14 days using a HOBOware U30 data-logger with 12-bit temperature sensors (Onset Computer Corporation, Bourne, MA).

## Quantification of animal behaviours

Fish behaviour was recorded by a centrally placed overhead infrared camera (Panasonic CCTV colour Camera, WV-CP620 with a 2X Lens variable focal WV-LZ61/2S) and lighting system. This allowed continuous digital video recording of fish movement through simulated day and night cycles. Videos were analyzed using Ethovision XT, Version 11 (Noldus, Wageningen, Netherlands) to quantify behaviours by automated animal tracking. Dynamic subtraction was used to target and track coordinates of each animal within the ATPT on a per second basis, and finished tracks were manually verified. The field of view was separated into eight zones corresponding to 16°C, 19°C (L, R), 21°C (L, R), 23°C (L, R), or 26°C (*Figure 1*). This was used to calculate fish preference for each thermal zone, velocity of movement, and migration between thermal zones. Data were compiled and used to calculate mean hourly temperature preference, velocity, and number of transitions across thermal zones.

## In vivo *Aeromonas* infection model and quantification of pathogen loads on skin wounds

*Aeromonas veronii* biovar sobria (NCBI Taxonomy ID: 114517) was previously isolated by our lab from goldfish cutaneous lesions (*Havixbeck et al., 2017*). For culture preparation, bacteria were inoculated into a 5-ml sterile trypticase soy media (BD Biosciences, Franklin Lakes, NJ) and cultured at room temperature on a shaker overnight. Fish were anesthetised in a tricaine methanesulfonate solution (02168510; Syndel, WA), a 4 × 4 patch of scales was removed, and minor abrasions made on the skin. The cutaneous wound was then inoculated with 10 μl of *A. veronii* log-phase culture broth (4.1 × 10$^8$ CFU/ml) before returning the fish to water. This inoculation dose was previously determined to promote a self-resolving acute inflammatory process where initial induction and subsequent resolution phases could be examined (*Havixbeck et al., 2017*). Infected fish were assigned randomly to different temperature categories, and they were also randomly selected at indicated time points for our experiments. Subsequent assessment of *A. veronii* numbers at the furuncle surface was used as a measure of pathogen load and shedding potential; surface bacteria were collected and plated onto 2× replicate tryptic soy agar plates. CFUs were quantified after 24-hr incubation at room temperature. Ketorolac (02162644; ATNAHS, Basildon, UK) administered intraperitoneally at 0.5 mg/kg of body weight was used in select experiments.

## Gene expression

Samples were collected, immediately frozen in liquid nitrogen, and stored at −80°C until use. Total RNA was extracted using TRIzol (15596026; Thermo Fisher Scientific, Waltham, MA) following the manufacturer's specifications. RNA concentration and quality were evaluated using a Nanodrop ND-1000 (Thermo Fisher Scientific, Waltham, MA) and Bioanalyser-2100 equipped with an RNA 6000 Nano Kit (5067-1511; Agilent Technologies, Santa Clara, CA). cDNA was synthesized using iScript Kit (1708891; BioRad, Mississauga, Canada) according to the manufacturer's specifications. qPCR was performed using the QuantStudio 6 Flex Real-Time PCR System (Applied Biosystems, Waltham, MA) where RQ values were normalized against gene expression on day 0 for each replicate time course and β-actin was used as a reference gene. Primers used are listed in *Supplementary file 1*. Relative quantification was performed according to the 2$^{-\Delta\Delta Ct}$ method.

## ROS production

Leukocyte isolations and ROS production evaluations were performed as previously described (*Havixbeck et al., 2016*; *Soliman et al., 2021*). After isolations, 500 μl of cell suspensions were incubated with 0.5 μl of CellROX Deep Red Reagent (C10491; Thermo Fisher Scientific, Waltham, MA) in the dark for 30 min to allow for cellular uptake. Cells were washed twice with 1× PBS$^{-/-}$ and fixed with 1% formaldehyde (47608; Sigma Aldrich, St. Louis, MO). Samples were centrifuged at 350 × *g* for 5 min at 4°C. Data were acquired using an ImageStream Mk II Imaging Flow Cytometer (Amnis, Seattle, WA), and analyzed via IDEAS Image Data Exploration and Analysis Software (Amnis, Seattle, WA). Cells were gated based on the normalized frequency of a fluorescent minus one sample.

## NO production

The production of NO was evaluated using 4-amino-5-methylamino-2',7'-difluorofluorescein diacetate (DAF-FM DA; D23844; Invitrogen, Waltham, MA). Following leukocyte isolation, 500 μl of cell

suspensions were incubated with DAF-FM DA at a concentration of 1 µM for 30 min in the dark. Cells were washed twice with 1× PBS$^{-/-}$ and fixed with 1% formaldehyde (47608; Sigma Aldrich, St. Louis, MO). Samples were centrifuged at 350 × $g$ for 5 min at 4°C. Data were acquired using an Image-Stream Mk II Imaging Flow Cytometer (Amnis, Seattle, WA), and analyzed via IDEAS Image Data Exploration and Analysis Software (Amnis, Seattle, WA). Cells were gated based on the normalized frequency of a fluorescent minus one sample.

## Prostaglandin E2 in goldfish plasma

Following cutaneous infection with *A. veronii*, fish were placed in either 16°C static, 26°C static, or dynamic fever thermal conditions. At 0, 12, 24, 48, and 72 hr post-infection, heparinized blood was collected from six individuals for each thermal condition and centrifuged at 2000 × $g$ for 10 min at 4°C. The resulting plasma supernatant was collected, aliquoted, and stored at −80°C until use. Using a Prostaglandin E2 ELISA Kit (514010; Cayman Chemical, Ann Arbor, MI), plasma samples were diluted 1:30 and PGE$_2$ protein concentrations were determined following the manufacturer protocol. Plates were read using the SpectraMax M2e plate reader (Molecular Devices, San Jose, CA) at 405 nm. Analysis and quantification of PGE$_2$ protein production were completed as per the manufacturer specifications (Cayman Chemical, Ann Arbor, MI).

## Histopathological analysis

Wound tissues were collected and fixed in 10% neutral-buffered formalin (SF98-4; Thermo Fisher Scientific, Waltham, MA). After processing tissues overnight in a series of ethanol, toluene, and wax washes using a Leica TP1020 benchtop tissue processor (Leica Biosystems, Concord, Canada), they were paraffin-embedded and sectioned (7 µm thickness) on slides using a Leica RM2125 RTS microtome (Leica Biosystems, Concord, Canada). Slides were deparaffinized and washed using two rounds of toluene (T324-1; Thermo Fisher Scientific, Waltham, MA) (5 min each) followed by rounds of 100%, 90%, 70%, and 50% ethanol (2 min each). For hematoxylin and eosin (H&E) staining, slides were placed in Surgipath Hematoxylin Gill III (3801542; Leica Biosystems, Concord, Canada) for 2 min, washed with running cold tap water for 15 min followed by 70% ethanol for 2 min, and then in Surgipath Eosin solution (3801602; Leica Biosystems, Concord, Canada) for 30 s. For Masson's Trichrome stain, slides were placed in hematoxylin Gill III for 1 min and washed with running cold tap water for 15 min. Slides were stained with ponceau-fuchsin (AC400211000; Thermo Fisher Scientific, Waltham, MA) for 2 min, rinsed in distilled water, differentiated in mordant in 1% phosphomolybdic acid (19400; Electron Microscope Sciences, Hatfield, PA) for 5 min. Slides were then stained with Aniline Blue solution (A967-25; Thermo Fisher Scientific, Waltham, MA) for 3 min and incubated in 1% phosphomolybdic acid then acetic acid solution (A38C-212; Thermo Fisher Scientific, Waltham, MA) for 5 and 3 min, respectively. Lastly, for both H&E and Masson's Trichrome, slides were dehydrated in series of alcohol, cleared in toluene, mounted with DPX Mountant (50-980-370; Thermo Fisher Scientific, Waltham, MA). Images were obtained using an AxioScope A1 microscope (Zeiss, Oberkochen, Germany).

## Statistics

Data were statistically analyzed and graphed using GraphPad v9.3.1 (San Diego, CA). Fish numbers used in each experiment were determined based on the minimal sample size ($n$) required to detect statistical significance, while considering animal ethics and care guidelines under the Canadian Council on Animal Care. Welch's $t$-test was used to determine if the means of two groups were significantly different. Data of both groups had a normally distributed population, but it was not assumed that they had the same variance. One-way analysis of variance (ANOVA) was used to compare the variance in the means of three or more categorical independent groups, considering a single independent factor or variable in the analysis. Data had a normally distributed population and each sample was drawn independently of the other samples. Additionally, the dependent variable was continuous. Two-way ANOVA was utilized for comparing the effect of two independent categorical factors on a dependent variable such as fold-change. The dependent variable was continuous and each sample was drawn independently of the other samples. Šídák's multiple comparisons test was used in select cases when a set of means were selected to compare across two groups and each comparison was assumed to be independent of the others. Mean values and correlations for behavioural data were calculated in Excel (Microsoft, Redmond, WA). R (version

3.3, The R Foundation for Statistical Computing, Vienna, Austria) was used to calculate multivariate statistics including principal component analysis (standard R package) and permutational multivariate ANOVA using distance matrices ('vegan' community ecology package). R code is included as part of the supplementary material.

## Acknowledgements

We thank M Reichert, C Gerla, J Edgington, M Axelsson, and J Johnston for help in the construction of the custom ATPT, the Science Animal Support Services for maintaining experimental fish, and A Shostak and Z Song for input on statistics and multivariate analyses. This work was supported by a Natural Sciences and Engineering Research Council of Canada grant to DRB (RGPIN-2018-05768). AMS and MEW were supported by Graduate Teaching Assistantships from the Department of Biological Sciences, University of Alberta. DT was supported by a CONICYT-Chile postdoctoral fellowship (Becas Chile N° 74170029). SLS was supported by a Natural Sciences and Engineering Research Council of Canada post-doctoral fellowship. The funders had no role in study design, data collection and analysis, decision to publish, or preparation of the manuscript.

## Additional information

### Funding

| Funder | Grant reference number | Author |
|---|---|---|
| Natural Sciences and Engineering Research Council of Canada | RGPIN-2018-05768 | Daniel R Barreda |
| Department of Biological Sciences, University of Alberta | Graduate Teaching Assistantships to AMS and MEW | Amro M Soliman Michael E Wong |
| Comisión Nacional de Investigación Científica y Tecnológica | Post-doctoral fellowship | Débora Torrealba |
| Natural Sciences and Engineering Research Council of Canada | Post-doctoral fellowship | Shawna L Semple |

The funders had no role in study design, data collection, and interpretation, or the decision to submit the work for publication.

### Author contributions

Farah Haddad, Amro M Soliman, Conceptualization, Data curation, Software, Formal analysis, Supervision, Funding acquisition, Validation, Investigation, Visualization, Methodology, Writing - original draft, Project administration, Writing - review and editing; Michael E Wong, Emilie H Albers, Shawna L Semple, Débora Torrealba, Ryan D Heimroth, Daniel R Barreda, Conceptualization, Resources, Formal analysis, Supervision, Funding acquisition, Investigation, Visualization, Methodology, Writing - original draft, Project administration, Writing - review and editing; Asif Nashiry, Software, Formal analysis, Visualization; Keith B Tierney, Conceptualization, Resources, Supervision

### Author ORCIDs

Shawna L Semple (ID) http://orcid.org/0000-0002-0176-1552
Daniel R Barreda (ID) http://orcid.org/0000-0003-4630-2840

### Ethics

All animals were maintained according to the guidelines of the Canadian Council on Animal Care. All protocols were approved by the University of Alberta Animal Care and Use Committee (ACUC-Biosciences protocols 706 and 355303). Fish were terminated via cervical dislocations using approved procedures. All efforts were made to minimize animal stress and to ensure that termination procedures were efficiently performed.

Decision letter and Author response
Decision letter https://doi.org/10.7554/eLife.83644.sa1
Author response https://doi.org/10.7554/eLife.83644.sa2

## Additional files

### Supplementary files
- MDAR checklist
- Supplementary file 1. Primer sequences for quantitative PCR.

### Data availability
All data generated during this study are included in the manuscript and supporting files. Source data have also been provided.

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
