## [Editor Report]

This study makes two important advances: First, the authors developed a new experimental system to study behavioral control of body temperature in fish. Second, using this experimental paradigm, the authors uncover the impact of body temperature regulation on immune defense and tissue repair. It presents important new insights into conserved defense mechanisms and as such the study would be of broad interest.

---

## [Decision Letter]

**Decision letter after peer review:**

Thank you for submitting your article "A cold-blooded vertebrate shows integration of antimicrobial defenses and tissue repair through fever" for consideration by *eLife*. Your article has been reviewed by 2 peer reviewers, and the evaluation has been overseen by a Reviewing Editor and Carla Rothlin as the Senior Editor. The following individual involved in review of your submission has agreed to reveal their identity: Martin F Flajnik (Reviewer #1).

*Reviewer #2 (Recommendations for the authors):*

The manuscript is very compelling with unique methods and combinations of approaches that support the overall conclusions made by the authors. I truly love the literature review and how the authors wrote the manuscript.

-The title could be a lot more effective and direct if the "model" part is removed. I suggest something like: "(Moderate) fever integrates/balances antimicrobial defense and tissue repair in teleosts".

- Line 152: byproduct not biproduct

- Lines 154-156 are not very clear but are very important to the overall conclusion of the manuscript. I would rephrase to something like: :fever in ectotherms favors early and selective induction of antimicrobial defenses coupled with controlled inflammation and accelerated wound repair overall acting as a fine-tuning mechanism of innate immunity to pathogens."

-Line 213: sickness behavior statement needs references

-Figure 3 and 5: gene names should blower case and capitalized. Also the Y axis should specified the fold change over what house keeping gene/s?

-Discussion: I am missing some comment linking back the discussion with the introduction in the context of the "moderate' fever model. How much more common is it in endotherms to display intermediate fever ranges compared to extreme? what is known between both types of fever in humans and how the data generated in the manuscript adds to the functional utility of intermediate fever states?

---

## [Author Response]

The reviewers have discussed their reviews with one another, and the Reviewing Editor has drafted this to help you prepare a revised submission.Reviewer #2 (Recommendations for the authors):The manuscript is very compelling with unique methods and combinations of approaches that support the overall conclusions made by the authors. I truly love the literature review and how the authors wrote the manuscript.

Thanks again. We really appreciate the comments.

-The title could be a lot more effective and direct if the "model" part is removed. I suggest something like: "(Moderate) fever integrates/balances antimicrobial defense and tissue repair in teleosts".

As suggested, we have modified the title to be more direct. It now reads: "Fever integrates antimicrobial defences, inflammation control, and tissue repair in a cold-blooded vertebrate".

- Line 152: byproduct not biproduct

Line has been revised (line 177 in updated manuscript).

- Lines 154-156 are not very clear but are very important to the overall conclusion of the manuscript. I would rephrase to something like: :fever in ectotherms favors early and selective induction of antimicrobial defenses coupled with controlled inflammation and accelerated wound repair overall acting as a fine-tuning mechanism of innate immunity to pathogens."

Line has been revised (line 177 in updated manuscript).

-Line 213: sickness behavior statement needs references

Reference now included (line 259 in updated manuscript).

-Figure 3 and 5: gene names should blower case and capitalized. Also the Y axis should specified the fold change over what house keeping gene/s?

Gene names in Figures 3, 4, 5 and Supplementary file 1 have been updated. Also, *actinb* is now listed as the house-keeping gene in figure legends where gene expression data is presented (refer to Figures 3, 4, 5 and Figure 3-figure supplement 1). This complements our previous description in the Methods (Gene expression section).

-Discussion: I am missing some comment linking back the discussion with the introduction in the context of the "moderate' fever model. How much more common is it in endotherms to display intermediate fever ranges compared to extreme? what is known between both types of fever in humans and how the data generated in the manuscript adds to the functional utility of intermediate fever states?

The prevalence of moderate forms of fever in humans compared to high-grade fever (>39^o^C) varies depending on the stimulus (e.g. type of infection), age of the affected individual, and factors such as underlying conditions. During the recent COVID-19 pandemic a prevalence of 82.49% versus 14.71% were reported for moderate and high-grade fevers, respectively (PLoS One. 2021; 16(4): e0249788). This, however, represents a rare opportunity for this comparison since under most common infections (e.g. seasonal influenza, common colds) moderate fevers are comparatively under-reported. Thus, we chose to avoid any specific commentary beyond those presented in lines 373-376 since these would undoubtedly misrepresent such prevalence in human populations. Numbers are even harder to capture accurately in other warm-blooded endotherms. However, we agree that added context would be beneficial. Thus, we now include the reference above as an example for the difference in prevalence of moderate and high-grade fevers under COVID-19 (line 544; reference 46).